# Predictive and Explanatory Uncertainties in Graph Neural Networks: A Case Study in Molecular Property Prediction

Marisa Wodrich[1], Aasa Feragen[1], and Mikkel N. Schmidt[*1]

[1]Technical University of Denmark, Kongens Lyngby, Denmark
{mawod, afhar, mnsc}@dtu.dk

## Abstract

Accurate molecular property prediction is a key challenge in fields such as drug discovery and materials science, where deep learning models offer promising solutions. However, the widespread use of these models is hindered by their lack of transparency and the difficulty in assessing the reliability of their predictions. In this study, we address these issues by integrating uncertainty quantification and explainable AI techniques to enhance the trustworthiness of graph neural networks for molecular property prediction. We focus on predicting two distinct properties: aqueous solubility and mutagenicity. By deriving explanations in the form of substructure attribution scores, we obtain interpretable explanations that signify which chemically meaningful substructures influence the model's predictions. We incorporate uncertainty quantification to evaluate the confidence of both the predictions and their explanations. Our results demonstrate that predictive uncertainty scores correlate with the accuracy of the predictions for both tasks. Uncertainties in the explanations also correlate with prediction correctness, and there is a weak to moderate correlation between the uncertainties in the predictions and those in the explanations. These findings highlight the potential of combining uncertainty quantification and explainability to improve the trustworthiness of molecular property prediction models.

## 1 Introduction

Molecular property prediction is a critical task in computational chemistry, material science, and drug discovery, where understanding the relationships between molecular structures and their properties can guide the discovery of new materials and therapeutics [1, 2]. Machine learning (ML), and particularly deep learning (DL) methods have revolutionized this field, enabling models to learn complex, high-dimensional representations of molecular data and provide accurate predictions for various molecular properties [3].

However, despite the promising performance of DL models in molecular property prediction, con-cerns about their reliability remains a significant barrier to their widespread adoption, particularly in high-stakes domains such as drug discovery and materials design. These models often lack transparency in how they arrive at predictions, which can be problematic in safety-critical applications. The absence of interpretability and reliability can make it difficult for chemists to trust model outputs and make informed decisions. Thus, ensuring the trustworthiness of predictions is a critical step toward advancing the utility of these models in real-world applications.

Explainable AI (XAI) techniques have emerged as a solution to address some of these challenges. By providing interpretable explanations for model predictions, XAI allows users to gain insights into the underlying decision-making process, fostering confidence in the predictions [4]. In the context of molecular property prediction, XAI can reveal how specific molecular substructures contribute to the model's output, providing valuable insights for researchers and guiding further experimental investigations. Additionally, uncertainty quantification (UQ) has become an essential tool in assessing the reliability of model predictions [5]. By quantifying the uncertainty associated with a prediction, UQ helps identify regions where the model is less confident and may be prone to errors, allowing for more reliable and cautious decision-making.

In this work, we aim to bridge these critical aspects of deep learning models in molecular property prediction: uncertainty quantification, explainability, and their interplay. Specifically, we

1. show how uncertainty quantification can be applied to molecular property predictions to assess their trustworthiness,

2. show how substructure explanations can be used to interpret these predictions, and propose and compare several ways to determine what role uncertainties play in these explanations,

3. show that there is a correlation between uncertainty scores and correctness of predictions for both predictive uncertainties and explanation uncertainties, and

4. analyze the relationship between these different uncertainties.

---

*Corresponding Author.

Proceedings of the 7th Northern Lights Deep Learning Conference (NLDL), PMLR 307, 2026.

Through these investigations, we aim to contribute to the development of more trustworthy and interpretable deep learning models for molecular property prediction.

## 2 Background

In recent years, ML and DL techniques have emerged as powerful tools for molecular property prediction [3]. These approaches, particularly neural networks, can capture complex patterns in molecular data and provide accurate predictions across a wide range of tasks, such as predicting solubility, toxicity, bioactivity, and mutagenicity. Graph neural networks (GNNs) have gained particular prominence as they operate directly on molecular graphs, representing atoms as nodes and bonds as edges, to reflect the molecular connectivity [6]. Despite the significant advances in predictive performance, challenges remain regarding the interpretability and reliability of deep learning models. These DL models offer little insight into the decision-making process, making it difficult to understand how certain molecular features contribute to the predicted properties.

In the context of molecular property prediction, XAI methods can help researchers understand which molecular characteristics, such as specific substructures, functional groups, or atom-level interactions, drive the predictions of a model. This interpretability is crucial for validating model predictions, especially in high-stakes domains such as drug discovery, where understanding the rationale behind a prediction can help researchers make more informed decisions and avoid potentially harmful or costly errors.

Various XAI techniques have been explored in molecular property prediction. These techniques include SHAP (SHapley Additive exPlanations) e.g. in [7], MolSHAP [8], LIME (Local Interpretable Model-agnostic Explanations), e.g. [9], and substructure-based explanation methods, e.g. [10]. The latter are especially interesting because they can reveal chemically meaningful substructures that drive the model's predictions. However, perturbation-based methods that mask parts of the input graph are not well suited for molecular property prediction. Metrics such as comprehensiveness [11], which measure how much predictions change when an explanation subgraph is removed, can in principle be optimized to find important substructures, but this stragegy can fail in molecular settings where even minor structural modifications can lead to large shifts in chemical properties. A better alternative is to apply substructure masking in a learned node representation that incorporates information from the full molecular graph [10], as well as rely on a fixed fragmentation scheme that ensures the resulting substructures remain connected and chemically meaningful. By identifying the key substructures, researchers can not only gain insight into the behavior of the model but also identify potential areas for further experimental validation or molecular optimization.

While XAI helps to understand model behavior, UQ provides a complementary approach to assess the reliability of predictions. UQ focuses on measuring the confidence or uncertainty in a model's outputs, offering valuable information on the regions where a model might be less certain or prone to error [5]. In molecular property prediction, the incorporation of uncertainty quantification can help identify predictions that are likely to be incorrect, thus improving the overall trustworthiness of the model, and can help determine which molecules should be selected for further experimental testing [12]. A variety of UQ methods exists that be can used for this task, including ensemble-based methods and distance-based methods.

Recent research has begun to explore the integration of XAI and UQ to provide a more comprehensive understanding of model reliability [13]. By integrating UQ with XAI, it becomes possible to evaluate not only how likely the prediction is to be correct and what the most plausible explanation for the model's output is, but also how reliable those explanations themselves are. This integrated approach has the potential to enhance the interpretability and trustworthiness of molecular property prediction models, offering a deeper understanding of the factors driving model decisions and their associated uncertainties. A gap persists in understanding how uncertainties in molecular property predictions connect to uncertainties in their corresponding explanations, which is essential for developing models that are reliable, interpretable, and actionable.

## 3 Methods

### 3.1 Data

Two datasets [10] are used in this study, one for predicting aqueous solubility (ESOL) and one for predicting mutagenicity. Each dataset was randomly split into training, test and validation data with a 80-10-10 split. Details about the data are shown in Table 1.

### 3.2 Model construction

The molecular prediction models are implemented as neural network ensembles [14], where each ensemble member is an independently trained relational graph convolution network (RGCN) model, following the design of Wu et al. [10] (see Fig. 1). Each model processes the molecular graph using three RGCN layers, after which attention pooling (a weighted sum of

**Table 1.** Datasets and their associated characteristics used in this study. Size indicates the total number of molecules in each dataset before splitting into training, test, and validation sets. Metric denotes the primary evaluation metric used for assessing the performance. The datasets do not include ground-truth explanation labels.

| Dataset | Task | Size | Metric |
|---|---|---|---|
| ESOL | Regression | 1111 | MSE |
| Mutagenicity | Classification | 7672 | AUC |

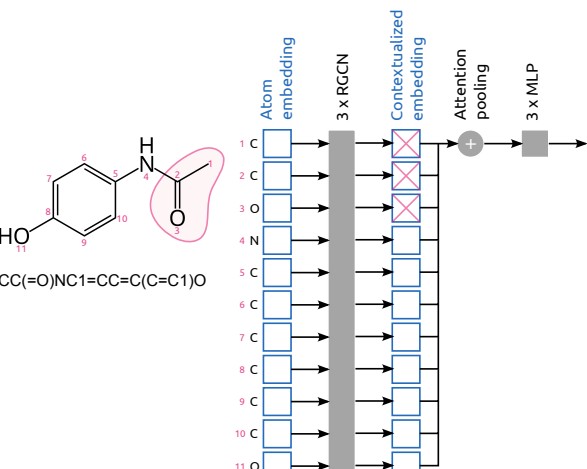

CC(=O)NC1=CC=C(C=C1)O

**Figure 1.** The model operates on molecular graphs using an initial atom embedding (and bond embedding, not illustrated) followed by three relational graph convolution (RGCN) layers that generate contextualized atom representations by message passing between the atoms. These representations are aggregated with an attention pooling layer and passed through three fully connected layers to produce the final prediction. When a fragment is masked, the contextualized embeddings of its atoms are set to zero while the atoms themselves remain in the graph during RCGN message passing. The full system uses an ensemble of ten independently trained models.

node features) produces a molecular-level embedding that is passed through three fully connected layers to generate the prediction. The ensemble members are initialized with different random seeds to ensure model diversity. The final prediction $F(x)$ for an input molecule $x$ is calculated as the average of the predictions across the ensemble members,

$$F(x) = \frac{1}{M} \sum_{m=1}^{M} f^{(m)}(x), \qquad (1)$$

where $M$ is the number of ensemble members and $f^{(m)}(x)$ is the prediction of ensemble member $m$.

### 3.3 Predictive uncertainty

Two different methods for quantifying predictive uncertainties are used in this study: The ensem-

ble variance (regression and classification) and the negative softmax score (only classification).

**Ensemble variance** The predictive uncertainty can be measured by the variance of the individual predictions across the ensemble. If the ensemble is certain that a prediction is correct, then all ensemble members should align in their predictions, while high variance means that the members disagree, and hence the overall ensemble is not sure. Formally, the predictive uncertainty is measured by

$$U_{\text{var.}}(x) = \frac{1}{M} \sum_{i=m}^{M} \left( F(x) - f^{(m)}(x) \right)^2. \qquad (2)$$

**Negative softmax score** For classification tasks, the predictive uncertainty can also be measured by using the output from the softmax activation function (after the last layer of the neural network) for the predicted class. Since high softmax scores means low uncertainty, the softmax values are negated to represent uncertainty instead of confidence. In the binary classification case, this is simply given as

$$U_{\text{softmax}}(x) = -\max \left( F(x), 1 - F(x) \right). \qquad (3)$$

### 3.4 Substructure explanations

We aim at explaining model predictions by assigning attribution scores to molecular fragments. We base our analysis on three different ways of breaking molecules into chemically meaningful substructures:

**BRICS** Molecules are split at retrosynthetically meaningful bonds, generating chemical substructures useful for design or combinatorial assembly [15].

**Murcko scaffold** Molecules are split into the central scaffold fragment (all rings and linkers) and the side-chain fragments attached to it, giving a natural division between core and substituents [16].

**Functional groups** Molecules are split into small fragments corresponding to known chemical groups linked to reactivity.

To assess how much a specific molecular substructure contributes to the model's prediction, we compare the full model prediction with the prediction obtained when that substructure is masked. Importantly, because masking molecular fragments in the input would create chemically unrealistic structures, we apply the masking to the final contextualized atom-wise representation of the model, immediately before the attention pooling. For each atom that is part of the masked substructure, the weight of its corresponding nodes are set to zero (see Fig. 1).

Based on these two different predictions, we can measure the signed attribution of this substructure, by contrasting the model output with and without masking. For a single model, we define the attribution as

$$a^{(m)}(x,s) = f^{(m)}(x) - f^{(m)}(x,s), \qquad (4)$$

where $f^{(m)}(x,s)$ is the prediction of the $m$-th ensemble member where the $s$-th substructure is masked. Similary we define the attribution for the ensemble prediction,

$$A(x,s) = \frac{1}{M} \sum_{m=1}^{M} a^{(m)}(x,s) \qquad (5)$$

$$= \underbrace{\frac{1}{M} \sum_{m=1}^{M} f^{(m)}(x)}_{F(x)} - \underbrace{\frac{1}{M} \sum_{m=1}^{M} f^{(m)}(x,s)}_{F(x,s)} \quad (6)$$

A value of $|A(x,s)|$ close to zero indicates that the masked substructure has little to no impact on the molecular property that is predicted. In contrast, a large absolute value indicates that this substructure is of high relevance.

## 3.5 Explanation uncertainty

The uncertainty of a substructure explanation can be measured by calculating each ensemble member's attribution score separately and taking the variance of this,

$$U(x,s) = \frac{1}{M} \sum_{m=1}^{M} \left( a^{(m)}(x,s) - A(x,s) \right)^2. \quad (7)$$

The overall uncertainty of the explanations across the entire molecule can be defined in different ways. Here, we will specifically look at four options.

1. The uncertainty of the molecular substructure with the strongest attribution,

$$U_{\text{highest}}(x) = U(x,s^*) \qquad (8)$$

where $s^* = \arg \max_s |A(x,s)|$.

2. The sum of uncertainty across all substructures,

$$U_{\text{all}}(x) = \sum_s U(x,s). \qquad (9)$$

3. The summed uncertainties weighted by the absolute value of the attribution,

$$U_{\text{weighted}}(x) = \sum_s |A(x,s)| \cdot U(x,s), \quad (10)$$

such that substructures with larger attribution contribute more to the overall uncertainty.

**Table 2.** Performance on the test datasets for predicting solubility and mutagenicty.

| Dataset | Metric | Result |
|---------|--------|--------|
| ESOL | MSE | 0.350 |
| Mutagenictiy | AUC | 0.896 |

4. The weigthed sum of uncertainties, where the weights are normalized across substructures,

$$U_{\text{scaled}}(x) = \frac{\sum_s |A(x,s)| \cdot U(x,s)}{\sum_s |A(x,s)|}, \qquad (11)$$

such that the metric is insensitive to the number of substructures.

## 3.6 Evaluation

The main performance metrics are the mean squared error (MSE) for predicting aqueous solubility on the ESOL dataset, and the area under the receiver operating characteristic curve (AUC) for predicting mutagenicity on the corresponding dataset. In all experiments we use $M = 10$ ensemble members.

To evaluate the preditive uncertainies, the relation between these uncertainties and the performance at molecular property prediction is studied. This is done by dividing the data into different subsets based on uncertainty scores and looking at the molecular property prediction for each of the subsets. Additionally, the performance on the test dataset is evaluated when excluding predictions with the highest uncertainty scores.

To evaluate the uncertainties of the explanations, the data is again divided into different subsets based on the explanation uncertainty, and the performance on each subset is calculated separately. This division into subsets is done separately for each explanation uncertainty method and for each of the three methods to determine substructure mask explanations.

The relation between the predictive uncertainties and the explanation uncertainties is evaluated by calculating the correlation coefficients for each combination of explanation uncertainty, predictive uncertainty, explanation method and dataset.

# 4 Results

## 4.1 Molecular property prediction

The performance of the RGCN ensemble for the task of predicting aqueous solubility and mutagenicity is shown in Table 2.

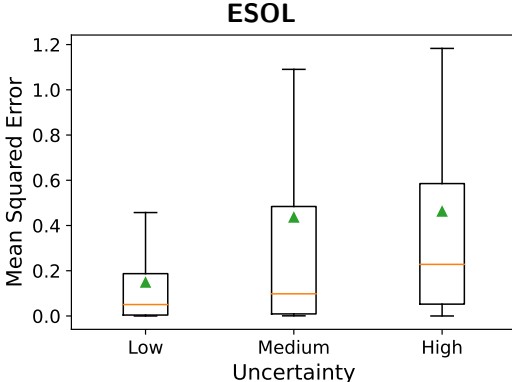

**Figure 2.** Mean squared error for predictions with low, medium and high predictive uncertainties for the ESOL test dataset. All subsets were chosen to be of equal size.

## 4.2 Predictive uncertainty

For both datasets a correlation between predictive uncertainties and correctness of the prediction was observed.

**ESOL** Dividing the ESOL test set into three equally sized subsets ranked by total predictive uncertainty shows a clear trend (see Fig 2): The lowest-uncertainty subset has the lowest MSE, the middle subset has an intermediate MSE, and the highest-uncertainty subset has the highest MSE on average. This is also reflected in Fig. 3: Applying an uncertainty threshold to select the most trustworthy predictions monotonically improves model performance, as reflected by a decreasing mean squared error (MSE) with more restrictive thresholds. That is, when considering only the molecules with the lowest predicted uncertainty, the MSE steadily drops.

**Mutagenicity** Figure 4 presents a similar analysis for the mutagenicity prediction task, showing accuracy and AUC for different uncertainty thresholds. The curves illustrate performance when high-uncertainty predictions are excluded and highlight the relationship between predicted uncertainty and accuracy, indicating that predictions with lower uncertainty are generally more reliable. The ensemble variance and the negative softmax score give similar results.

## 4.3 Explanations

An example of several substructure mask explanations for a molecule from the Mutagenicity dataset is shown in Fig. 5, using all three methods for dividing the molecule into meaningful chemical substructures (BRICS, Murcko scaffolds and Functional groups).

The performance on test data falling into four brackets of explanation uncertainty from low to high is shown in Fig. 6 (ESOL) and Fig. 7 (Mutagenicity).

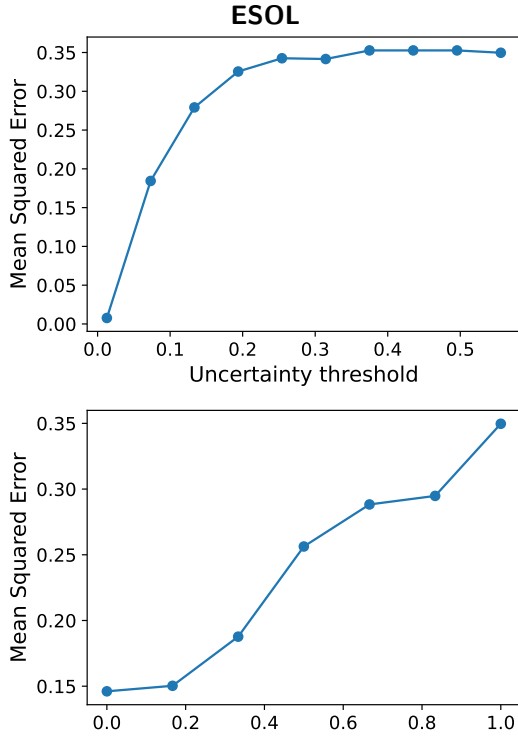

**Figure 3.** Mean squared error on ESOL test data when only including data points with the lowest predictive uncertainties. The upper row shows the performance based on uncertainty threshold values, while lower row shows the performance based on the fraction of included data.

For both datasets, there is a general correspondence between explanation uncertainty and model performance, such that high molecular property prediction performance is achieved on data with low explanation uncertainty for all four explanation uncertainty metrics across all types of chemical substructures (BRICS, Murcko scaffolds, Functional groups), with the exception of Functional groups for ESOL, where no significant relation between the explanation uncertainies and the MSE of the prediction could be observed.

## 4.4 Relation of predictive uncertainty and explanation uncertainty

**ESOL** On the ESOL dataset, weak to moderate correlations were found between the predictive uncertainties and the different types of explanation uncertainties (see Table 3). The strongest correlation was found for BRICS substructures using the $U_{\text{highest}}$ explanation uncertainty.

**Mutagenicity** The correlation between explanation uncertainties and predictive uncertainties for the mutagenicity dataset were found to be moderate to strong (see Table 3 for the results when using the

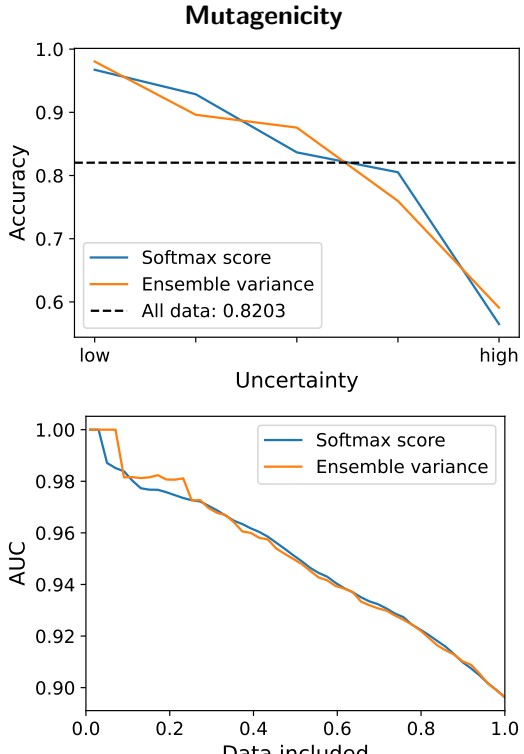

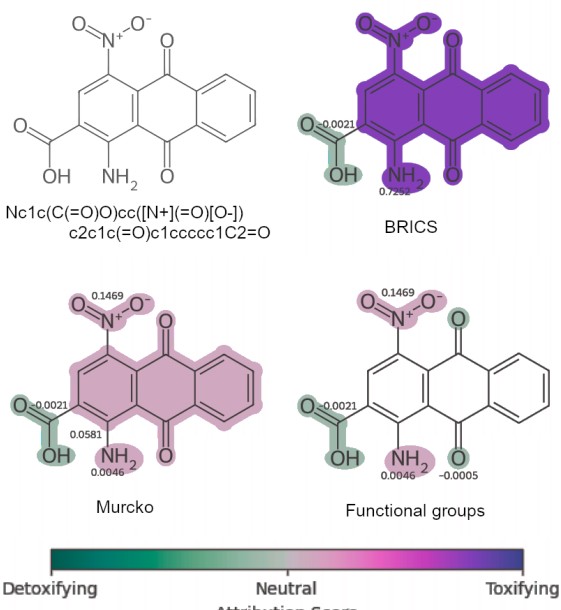

**Figure 5.** Substructure mask explanations for an example molecule from the Mutagenicity dataset. Purple color refers to a positive impact (toxifying) of a substructure to the final prediction, and green color refers to a negative impact (detoxifying). The color of the substructure indicates its effect strength, with the attribution value shown next to it.

**Figure 4.** Performance on Mutagenicity test data when only including data points with the lowest predictive uncertainties. The upper figure shows the accuracy for different uncertainty thresholds, and the lower figure shows the AUC for different fractions of included data.

ensemble variance or the softmax score as the predictive UQ method). Similar to ESOL, the BRICS substructure explanation uncertainties correlate the most with the predictive uncertainties, however here, the highest correlations are found when using $U_{\text{all}}$ explanation uncertainty.

Across all preditive UQ methods, all four explanation UQ methods and both datasets, the functional groups explanation uncertainties showed the lowest correlation scores. An example scatter plot of the relation between predictive uncertainties and explanation uncertainties is shown in Fig. 8.

# 5   Discussion

One key observation of this study is that predictive uncertainty scores correlate with the accuracy of predictions, supporting the hypothesis that uncertainty estimation can be a reliable indicator of prediction correctness. As shown in Fig. 3 for ESOL and in Fig. 4 for Mutagenicity, samples with high predictive uncertainties can be excluded, which will lead to an improved overall performance of the remaining data. Although using uncertainty thresholds like this comes at the expense of not being able to make predictions for some molecules, it also has

the benefit that the remaining predictions are more trustworthy and reliable, which can be crucial in high-stake applications such as drug design.

By dividing the data into different subsets based on their predictive uncertainties, this observation could be further confirmed. For ESOL, the MSE for low-uncertainty predictions was close to zero with a median of 0.05 (Fig. 2). For Mutagenicity, the likelihood of a low-uncertainty prediction being correct was close to 1.0, while the likelihood of high-uncertainty prediction was just above chance level (Fig. 4). This is a strong indicator for the argument that predictions with high uncertainties should not be blindly trusted, as there is a high chance that they are incorrect.

Furthermore, explanation uncertainties were shown to have a correlation with prediction correctness, emphasizing that they are not only of importance for increasing model interpretability, but that when the explanation is unsure or unclear, it is often connected to an unreliable prediction. Although this relation is clear, it is less strong than the relationship between the predictive uncertainty and the performance, which can be seen in Fig. 6 and 7. For mutagenicity, the accuracy of predictions for samples with low explanation uncertainties was around 0.87-0.95, and around 0.70-0.80 for samples with high explanation uncertainties, which is still a large performance difference, suggesting that predic-

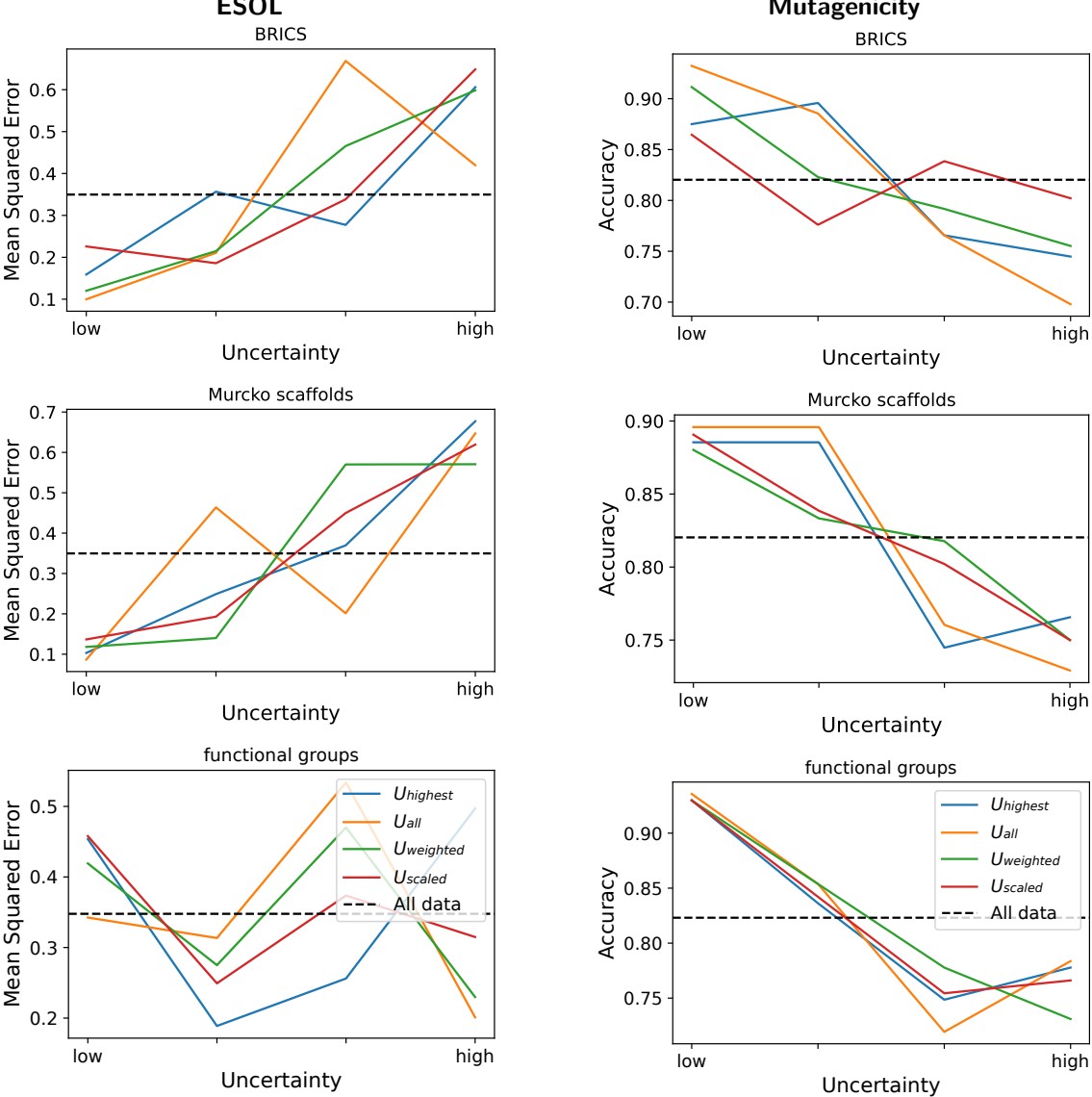

**ESOL**

**Mutagenicity**

**Figure 6.** Mean squared error for predictions with low to high explanation uncertainties on the ESOL test dataset.

**Figure 7.** Likelihood of a prediction being correct given the uncertainty of the explanation for the Mutagenicity test dataset.

tions should not be trusted when the explanation uncertainty is high.

The findings indicate that predictive uncertainty and explanation uncertainty are interrelated, but the strength of this relationship varies depending on the dataset and uncertainty estimation method used. In the ESOL dataset, weak to moderate correlations were observed, with BRICS explanations showing the strongest alignment with predictive uncertainty (Table 3). In contrast, the mutagenicity dataset exhibited moderate to strong correlations, particularly when using ensemble variance as the predictive UQ method (Table 3).

The strongest correlations were observed when summing the attribution scores of all possible explanations within a single substructure explanation method. While BRICS and Murcko scaffold explana-

tions demonstrated the highest correlation with predictive uncertainty, functional group explanations consistently showed lower correlation scores.

When comparing the UQ methods in this study, the variance-based ensemble uncertainty and the softmax score behave similarly in terms of predictive performance, as measured by AUC and accuracy (Fig. 4). However, regarding correlation with predictive uncertainty, the ensemble variance shows a much stronger relationship. The four different explanation uncertainty methods all exhibited similar performance at detecting untrustworthy predictions, but they differ largely in their correlations with the predictive uncertainties.

Since it was shown that both the predictive uncertainties and the explanation uncertainties have a strong relation of their scores to the correctness

**Table 3.** Correlations between predictive uncertainty and the different types of explanation uncertainties.

| | | BRICS | Murkco scaffold | Functional groups |
|---|---|---|---|---|
| ESOL | $U_{\text{highest}}$ | **0.45** | **0.37** | **0.17** |
| | $U_{\text{all}}$ | 0.14 | 0.34 | 0.07 |
| | $U_{\text{weighted}}$ | 0.06 | 0.29 | -0.05 |
| | $U_{\text{scaled}}$ | 0.35 | 0.44 | 0.15 |
| Mutagen. (ensemble) | $U_{\text{highest}}$ | 0.47 | 0.59 | 0.29 |
| | $U_{\text{all}}$ | **0.76** | **0.66** | **0.40** |
| | $U_{\text{weighted}}$ | 0.5 | 0.43 | 0.18 |
| | $U_{\text{scaled}}$ | 0.24 | 0.52 | 0.25 |
| Mutagen. (softmax) | $U_{\text{highest}}$ | 0.24 | 0.36 | 0.13 |
| | $U_{\text{all}}$ | **0.57** | **0.46** | **0.24** |
| | $U_{\text{weighted}}$ | 0.24 | 0.2 | 0.03 |
| | $U_{\text{scaled}}$ | -0.03 | 0.27 | 0.10 |

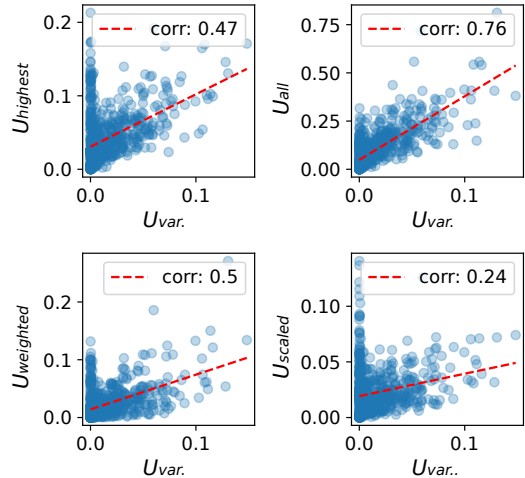

**Figure 8.** Correlation between predictive uncertainty and explanation uncertainty from the BRICS substructure explanations for the Mutagenicity test datset. Here, the predictive uncertainty was measured as the ensemble variance.

of a prediction, but the correlations between these uncertainties are mostly only moderate (with a few being high or low), these results suggest that predictive uncertainty and explanation uncertainty provide complementary perspectives on model trustworthiness. The correlations, varying in strength, indicate that combining both approaches may yield a more comprehensive measure of reliability.

# 6 Conclusion

This study shows that UQ can enhance a model's trustworthiness by identifying predictions that are likely unreliable, and that explainability methods offer complementary insight into how the model arrives at its predictions. For both datasets, a clear correlation between predictive uncertainties and correctness of the prediction was observed. A similar relation was found between the explanation uncertainties and the correctness of the prediction. When comparing the predictive uncertainties and the explanation uncertainties, positive correlations were found, with the strongest one when using the sum of the attribution scores of all possible explanations for a molecule within one substructure explanation method as the measure for explanation uncertainty. While some correlations were only weak, the results suggest that the different methods find different data samples untrustworthy, implying that a combination of all methods should be used to decide whether to trust a prediction or not.

**Limitations and future work** In future work, it should be explored how the different uncertainty measures could be combined into one meaningful, potentially more powerful, measure of trustworthiness. This could be done by training a small neural network that takes all the uncertainty scores and the prediction as input, and learns to predict whether this prediction was correct.

Additionally, more uncertainty quantification methods for measuring the predictive uncertainties should be taken into account. Here, it might be interesting to also study methods that decompose the total predictive uncertainy into its epistemic and aleatory parts.

More advanced graph neural networks that take into account the geometry of the molecules could also be explored in the future.

The explainability framework used in this work is based on masking in the contextualized atom embeddings of a graph neural network after message passing. A possible failure mode is that global molecular properties could be encoded not in the responsible atoms or fragments, but in structurally central substructures, leading to incorrect attribution. While this is less likely when the training data cover a broad range of chemical diversity, this should be examined in detail, although that is difficult without access to fragment-level ground truth attribution labels.

Furthermore, this study has the limitation of only investigating two datasets. This should be extended to more different datasets, with different molecular property prediction tasks.

# Acknowledgments

This work was supported by the Novo Nordisk Foundation under grant no NNF22OC0076658 (Bayesian neural networks for molecular discovery) and NNF20OC0062606 (Center for basic machine learning research in life science).

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
